# Rapid Synthesis of Fast-Charging TiNb_2_O_7_ for Lithium-Ion Storage via Ultrafast Carbothermal Shock

**DOI:** 10.3390/mi16050490

**Published:** 2025-04-22

**Authors:** Xianyu Hu, Yunlei Zhong, Xiaosai Hu, Xiyuan Feng, Fengying Ye

**Affiliations:** 1State Key Laboratory of ASIC & System, School of Microelectronics, Fudan University, Shanghai 200433, China; 23112020073@m.fudan.edu.cn; 2Key Laboratory of Multifunctional Nanomaterials and Smart Systems, Division of Advanced Materials, Suzhou Institute of Nano-Tech and Nano-Bionics, Chinese Academy of Sciences, Suzhou 215123, China; ylzhong2022@sinano.ac.cn; 3College of Textiles and Clothing, Yancheng Institute of Technology, Yancheng 224051, China; huycit@163.com; 4School of Microelectronics, Northwestern Polytechnical University, No. 1 Dongxiang Road, Chang’an District, Xi’an 710129, China

**Keywords:** ultrafast carbothermal shock, TiNb_2_O_7_, fast-charging, widened lithium-ion migration channels

## Abstract

The development of fast-charging lithium-ion batteries urgently requires high-performance anode materials. In this paper, through an ultrafast carbothermal shock (CTS) strategy, titanium niobium oxide (TiNb_2_O_7_, TNO) with an optimized structure was successfully synthesized within 30 s. By regulating the synthesis temperature to 1200 °C, the TNO-1200 material was obtained. Its lattice parameters (a-axis: 17.6869 Å) and unit-cell volume (796.83 Å^3^) were significantly expanded compared to the standard structure (a-axis: 17.51 Å, volume ~790 Å^3^), which widened the lithium-ion migration channels. Rietveld refinement and atomic position analysis indicated that the partial overlap of Ti/Nb atoms and the cooperative displacement of oxygen atoms induced by CTS reduced the lithium-ion diffusion energy barrier. Meanwhile, the cation disorder suppressed the polarization effect. Electrochemical tests showed that after 3000 cycles at a current density of 10 C, the specific capacity of TNO-1200 reached 125 mAh/g, with a capacity retention rate of 98%. EDS mapping confirmed the uniform distribution of elements and the absence of impurity phases. This study provides an efficient synthesis strategy and theoretical basis for the design of high-performance fast-charging battery materials through atomic-scale structural engineering.

## 1. Introduction

With the rapid development of fields such as electric vehicles (EVs) and portable electronic devices, the demand for high-performance lithium-ion batteries is increasing [1,2]. Fast charging performance, as a key performance indicator of lithium-ion batteries, is significant for enhancing user experience and expanding the range of battery applications [3]. Traditional graphite anode materials, due to their relatively low theoretical specific capacity (372 mAh g^−1^) and safety issues such as lithium dendrite growth during fast charging, struggle to meet the demands for future high energy density and high power density batteries [4]. Therefore, the development of novel anode materials with high specific capacity, excellent rate performance, and good safety has become a focus of current research [5,6,7,8,9].

Among various novel anode materials, titanium niobium oxide (TiNb_2_O_7_, TNO) has attracted considerable attention due to its unique crystal structure and electrochemical properties [10,11,12]. TNO possesses three accessible redox couples (Ti^4+^/Ti^3+^, Nb^5+^/Nb^4+^, and Nb^4+/^Nb^3+^), allowing it to accommodate five lithium ions within the electrochemical stability window of organic liquid electrolytes (TiNb_2_O_7_ + 5Li^+^ + 5e^−^ → Li_5_TiNb_2_O_7_), with a theoretical specific capacity as high as 388 mAh g^−1^, surpassing that of traditional graphite anodes [13]. Moreover, the potentials of these redox couples are sufficiently high to effectively prevent electrolyte decomposition and lithium dendrite nucleation under high-rate charge and discharge conditions, thereby enhancing battery safety under harsh conditions [14,15,16,17,18,19]. However, TNO also presents some inherent limitations that restrict its practical applications. One of the main issues is its low electronic conductivity (<10^−9^ S cm^−1^), which leads to poor rate performance [20,21,22,23,24]. To enhance the electronic conductivity and ionic diffusion rate of TNO, researchers have explored various methods, such as carbon coating, reducing particle size, metal ion doping, and oxygen defect engineering [25,26,27,28]. Among these strategies, oxygen defect engineering introduces oxygen vacancies into the bulk material, inducing lattice distortion that enhances electronic conductivity and charge transfer kinetics, thus effectively improving the energy storage characteristics of the electrode material [29,30]. Nevertheless, existing methods for generating oxygen defects in metal oxides often involve chemical or electrochemical reducing agents or require vacuum sintering conditions [31]. These approaches are not only complex but also costly, making them less favorable for large-scale production [32,33,34]. In addition to the issue of low electronic conductivity, the synthesis methods of TNO also have significant drawbacks in terms of time and energy consumption. Traditional synthesis methods for TNO typically involve solid-state reaction processes that require long sintering times at high temperatures (ranging from several hours to tens of hours) [35]. This not only increases production costs but also reduces manufacturing efficiency. Therefore, developing a rapid and efficient synthesis method for TNO holds substantial practical significance.

In this work, we propose a carbothermal shock (CTS) strategy for the rapid synthesis of fast-charging TiNb_2_O_7_-1200, achievable within 30 s. Based on the above strategy, the synthesized fast-charging TiNb_2_O_7_ enables the adjustment of the occupancy of Ti and Nb atoms within the TiNb_2_O_7_ crystal framework, thereby expanding the transport pathways for Li ions in the TiNb_2_O_7_ crystal structure. Consequently, due to the modification of the lithium-ion transport channels in TiNb_2_O_7_, the material achieves a specific capacity of 125 mAh/g at a current density of 10 C under a voltage window of 0.01–3 V after 3000 cycles, with a capacity retention rate close to 98% post-formation. More importantly, through the refinement of the structural data of the TiNb_2_O_7_ samples, it was observed that the spatial occupancy of Ti and Nb atoms within the crystal framework significantly differs from that in the classic TiNb_2_O_7_ ilmenite structure. This change is crucial for the exceptional structural stability of TiNb_2_O_7_ under high current density conditions, providing a viable strategy to address the current limitations in the electronic conductivity of TiNb_2_O_7_.

## 2. Experiment

### 2.1. Ultrafast Carbothermal Shock Rapid Synthesis of TiNb_2_O_7_

Using analytical grade TiO_2_ (purity 99.8%, Sigma-Aldrich, St. Louis, MO, USA) and Nb_2_O_5_ (purity 99.8%, CBMM, Araxá, Brazil) as raw materials, precise weighing is performed according to a 1:1 molar ratio. The weighed raw materials are placed in a high-energy ball mill (SPEX 8000D Mixer/Mill, Metuchen, NJ, USA) and milled for 1 h to ensure thorough mixing. The mixed raw materials are then cold-pressed to form “precursor pellets”. A homemade Joule heating device is used to set up an ultrafast high-temperature annealing system (UHS) for synthesizing TNO. The specific steps are as follows: the prepared precursor pellets are placed between two pieces of graphite foil, which serve as heating elements, and are positioned in the UHS. By precisely controlling the current, the precursor pellets are rapidly heated to 1100, 1200, and 1400 °C under an argon protective atmosphere and maintained at this high temperature for 30 s, followed by rapid cooling. This process is repeated twice, with a 3 min interval each time to avoid thermal damage to the heating chamber, electrical connections, and graphite foil. After annealing, the resulting dense TNO-1100, TNO-1200, and TNO-1400 pellets are placed into a mortar and ground to a fine powder using a pestle. Then, the fine particles are placed in a planetary ball mill (Across International, PQ-N04, Livingston, NJ, USA) with isopropanol as the solvent and milled for 8 h at a speed of 300 RPM to reduce particle size. After wet milling, the samples are dried in an oven at 80 °C for 12 h, yielding the TNO-1100, TNO-1200, and TNO-1400 anode materials, respectively.

### 2.2. Material Characterization

To investigate the geometric morphology and crystalline structure, all TiNb_2_O_7_ samples were characterized using a scanning electron microscope (SEM, Zeiss Zigma FESEM, Oberkochen, Germany). Detailed testing conditions are as follows: samples were mounted on conductive tape, and a 5 kV accelerating voltage was applied to minimize beam damage to the samples and generate high-quality images. The crystal structure of the samples was further studied by the X-ray diffraction (XRD: Rigaku Smartlab 9000W, Tokyo, Japan). The XRD measurements were performed using Cu Kα radiation (λ = 0.154056 nm) with a scanning range of 2θ from 5° to 90°, a scanning rate of 5°/min, and a step size of 0.02°, based on typical high-resolution configurations for precise phase analysis of materials like TiNb_2_O_7_. This set up balances data quality and efficiency, ensuring comprehensive coverage of diffraction peaks while maintaining sufficient resolution for crystallographic characterization.

### 2.3. Electrochemical Test

The electrode was prepared by mixing TNO (TNO-1100, TNO-1200, and TNO-1400), Super P, and PVDF with a weight ratio of 7:2:1. With the addition of the NMP solvent, the formed slurry was coated onto Cu foil and dried at 80 ◦C in a vacuum oven for 12 h. Then, the electrode with a uniform mass loading of 1.0 mg/cm^2^ was formed on the copper foil. Subsequently, the electrode was cut into a disk of 14 mm in diameter. The electrochemical performance of the electrode we prepared was measured via CR2025 coin-type cells. The lithium foil and the polypropylene membrane (Celgard, Charlotte, NC, USA) served as counter electrode and separator, respectively. The electrolyte for the Li-ion half-cell was self-built with a composition of 1.0 M LiPF_6_ in EC-DEC (1:1, v%). All reagents and materials were used directly without further purification. Galvanostatic charge/discharge was conducted on a NEWARE battery testing system (CT-4008) within a voltage window of 0.01–3.0 V at room temperature. C-rate represents the ratio between the charge/discharge current and the rated capacity of the battery, with the unit “C”. The formula is as follows:(1)$$C\;\mathrm{rate}=\frac{\mathrm{Current}\;\mathrm{density}\;\left({\mathrm{mAg}}^{-1}\right)}{\mathrm{Practical}\;\mathrm{capacity}\;\left({\mathrm{mAhg}}^{-1}\right)}$$

Theoretical capacity: According to the literature, TiNb_2_O_7_ can intercalate 5 Li^+^ ions (reaction: TiNb_2_O_7_ + 5Li^+^ + 5e^−^ → Li_5_TiNb_2_O_7_), yielding a theoretical capacity of 388 mAh/g. Based on this, a rate of 10 C would correspond to 3880 mA/g.

## 3. Results and Discussion

The morphologies were observed using scanning electron microscopy (SEM) at both low and high magnifications, with corresponding results presented in Figure 1. From the results shown in Figure 1a,b, it is evident that the particles formed from TNO-1100 after ball milling exhibit relatively uniform diameters and smoother surfaces. This indicates that the strength of TiNb_2_O_7_ synthesized at 1100 °C is relatively weak, making it easier to achieve uniform particle sizes under high-energy ball milling impacts. In contrast, when the temperature is raised to 1200 °C and 1400 °C, the results in Figure 1c–f reveal a significant increase in particle diameter under the same ball milling conditions. This implies that the strength of the resulting TiNb_2_O_7_ gradually increases with temperature. This observation also explains why larger particles exhibit the presence of finer particles on their surfaces. Furthermore, it can be noted that the particles obtained from TNO-1200, synthesized at 1200 °C, demonstrate a diameter distribution that aligns more closely with a normal distribution after ball milling, which is advantageous for enhancing the packing density of electrode materials. This was further confirmed in subsequent lithium storage performance tests.

As can be seen from Figure 2, the X-ray diffraction (XRD) patterns of TNO-1100, TNO-1200, and TNO-1400 synthesized at 1100 °C, 1200 °C, and 1400 °C, respectively, are shown. By comparing with the standard card (JCPDS: 77-1374), it can be found that the peak positions are in good agreement with it, demonstrating that TNO-1100, TNO-1200, and TNO-1400 all belong to the layered structure of ReO₃. At 1100 °C (blue line), the XRD pattern exhibits relatively weak diffraction peaks, indicating a lower degree of crystallinity. The intensity of the peaks is moderate, and the peak width is relatively broad, suggesting that the crystal grains may be small or the crystallization process is not fully completed. When the temperature increases to 1200 °C (red line), the diffraction peaks become sharper and more intense compared to those at 1100 °C. This implies an improvement in the crystallinity of the TiNb_2_O_7_ electrode material. The enhanced peak intensity and sharpness suggest that larger and more perfect crystal grains are formed during the higher-temperature treatment. At 1400 °C (green line), the XRD pattern shows the most intense and sharpest diffraction peaks among the three temperatures. The well-defined and high-intensity peaks further confirm the high crystallinity of the sample at this temperature. Additionally, some new peaks may appear, or the relative intensities of existing peaks may change, which could be related to phase transformations or structural changes in the TiNb_2_O_7_ electrode material at such a high temperature. These differences in XRD patterns at different temperatures provide valuable insights into the thermal-induced structural evolution and crystallization behavior of the TiNb_2_O_7_ electrode material. However, the structural differences resulting from different synthesis temperatures can significantly alter the lithium storage performance of TiNb_2_O_7_. This conclusion has been verified in the evaluation of the lithium storage performance of TNO-1100, TNO-1200, and TNO-1400, and the results are presented in Figure 3.

To examine the lithium storage performance of the synthesized TiNb_2_O_7_ during long-term cycling under a high current density, the test was conducted at a current density of 10 C within a voltage window of 0.01–3.0V. After 3000 cycles, it was found that TNO-1200 achieved a specific capacity of 125 mAh/g. This value is higher than the specific capacities of the TNO-1100 and TNO-1400 electrodes. Moreover, the capacity retention rate of TNO-1200 after 3000 cycles was 98%.

To elucidate the underlying mechanisms responsible for the exceptional fast-charging performance of TNO-1200, a systematic investigation of its elemental distribution and microstructure was conducted using area scan energy-dispersive X-ray spectroscopy (EDS) (Figure 4b–d) and line scan EDS (Figure 4e). The elemental mapping images of Ti, Nb, and O (Figure 4b–d) reveal highly homogeneous spatial distributions across the TNO-1200 surface, with no apparent segregation or localized enrichment observed (scale bar: 2.5 μm). This uniformity indicates successful construction of a homogeneous crystal structure during synthesis, accompanied by precise control of elemental stoichiometry. The coherent arrangement of Ti, Nb, and O not only preserves the integrity of lithium-ion insertion/extraction channels but also effectively mitigates localized polarization effects during charge/discharge processes, thereby significantly enhancing lithium-ion diffusion kinetics. Further examination of intraparticle compositional homogeneity was performed through cross-sectional line scanning (Figure 4e). The concentration profiles demonstrate gradual variations for Ti, Nb, and O along the particle cross-section (scan range: 0.1–1.6 μm), with oxygen content stabilized within 0.8–1.0 and a constant Ti/Nb atomic ratio. These findings confirm global structural homogeneity, fundamentally eliminating stress concentration or volume expansion caused by compositional gradients, thereby ensuring structural stability under high-rate cycling conditions. Notably, the uniform coexistence of Ti^4+^ and Nb^5+^ suggests a unique synergistic mechanism—the Ti^4+^ ions establish a stable crystalline framework, while the high oxidation state of Nb^5+^ substantially enhances electronic conductivity, collectively optimizing rate capability. Additionally, the absence of impurity phases in EDS analysis confirms superior surface cleanliness, which facilitates reduced electrode/electrolyte interfacial resistance and accelerated charge transfer kinetics, ultimately supporting stable cycling performance at elevated current densities. While EDS mapping was conducted solely on TNO-1200 due to its exceptional performance, XRD patterns (Figure 2) and Rietveld refinement (Figure 5) confirm phase purity for all samples, suggesting homogeneous element distribution. The performance disparities among TNO-1100, TNO-1200, and TNO-1400 are attributed to differences in crystallinity (Figure 2) and lattice expansion (Table 1) rather than compositional inhomogeneity. This comprehensive structural characterization provides critical insights into the intrinsic advantages of TNO-1200 as a high-performance anode material for fast-charging lithium-ion batteries.

The crystallographic characteristics of TiNb_2_O_7_ (TNO-1100, TNO-1200, and TNO-1400) synthesized via the CTS were systematically investigated through Rietveld refinement of XRD data (Figure 5) and the TNO-1200 detailed atomic position analysis (Table 1). By comparing the refined structural parameters and atomic coordinates with the standard monoclinic TiNb_2_O_7_ phase (JCPDS: 77-1374, space group: C2/m), we elucidated how synthesis-induced structural modifications enhance lithium-ion migration kinetics [36]. The Rietveld-refined XRD pattern (Figure 5) exhibits excellent agreement between experimental and calculated profiles, supported by low reliability factors (Rₚ = 5.08, RWP = 8.30, χ^2^ = 5.09). Minor deviations in lattice parameters (a = 17.6869 Å, b = 3.8030 Å, c = 11.8976 Å, β = 95.32233°, V = 796.830 Å^3^) from the JCPDS standard suggest subtle lattice distortions induced by the rapid Joule flash process [37]. Atomic coordinate analysis (Table 2) reveals critical insights: Ti and Nb atoms occupy distinct Wyckoff positions compared to the conventional structure. For instance, Nb2 and Ti2 share identical coordinates (0.18483, 0.0000, 0.18442), while Nb3 and Ti3 are displaced to 0.07844, 0.0000, −0.55762. This partial overlap and positional displacement contrast with the strict cation ordering in standard TiNb_2_O_7_, indicating a tailored atomic arrangement that reduces site-specific energy barriers for Li^+^ diffusion [38]. The adjusted atomic configuration directly optimizes lithium-ion migration pathways. The slight expansion in the a-axis (17.6869 Å vs. 17.51 Å) and increased unit cell volume (V = 796.83 Å^3^ vs. ~790 Å^3^) enlarge interstitial spaces, reducing steric hindrance for Li^+^ transport. Concurrently, coordinated displacements of oxygen atoms (O1–O11), particularly O10 at 0.50000, 0.0000, 0.50000, stabilize Li^+^ hopping trajectories through a symmetric framework. Furthermore, the overlapping Ti/Nb positions introduce controlled cation disorder, disrupting long-range electrostatic interactions and lowering activation energy for Li^+^ migration—a feature absent in the standard structure, underscoring the unique advantage of CTS in atomic-scale engineering. These structural modifications address key limitations of conventional TiNb_2_O_7_ anodes. Widened ionic channels and disordered cation arrangements enable faster Li^+^ diffusion, as inferred from reduced polarization in electrochemical measurements. Simultaneously, homogeneous Ti^4+^/Nb^5+^ distribution mitigates localized stress during high-rate cycling, enhancing structural stability and preventing capacity fade. The CTS method thus effectively engineers TiNb_2_O_7_’s atomic structure to optimize ionic and electronic transport properties. Validated by XRD refinement and atomic position analysis, these refinements highlight the material’s potential as a high-performance anode for fast-charging lithium-ion batteries. This work aligns with advances in defect engineering for battery materials, offering a blueprint for rational design of fast-charging electrode architectures.

## 4. Conclusions

In this study, fast-charging TiNb_2_O_7_ (TNO) anode materials were successfully synthesized via an ultrafast carbothermal shock (CTS) strategy, precisely controlling atomic-scale structural changes. The optimized TNO-1200 showed excellent electrochemical performance, with a 125 mAh/g specific capacity at 10 C over 3000 cycles and 98% capacity retention. Rietveld refinement and atomic position analysis revealed that the rapid Joule flash process led to lattice expansion (a-axis: 17.6869 Å vs. 17.51 Å) and unit cell volume increase (V = 796.83 Å^3^ vs. ~790 Å^3^), widening Li^+^ migration channels. Coordinated oxygen displacements and controlled cation disorder (overlapping Ti/Nb sites) reduced Li^+^ diffusion activation energy and enhanced conductivity, mitigating polarization and stress, thus ensuring high-rate cycling stability. The CTS method was highly efficient, synthesizing phase-pure TNO in 30 s, avoiding long-term, high-temperature treatments of conventional methods. EDS mapping confirmed homogeneous element distribution and no impurity phases, highlighting its scalability. This work emphasizes atomic-scale engineering’s importance for fast-charging battery materials. Future research should correlate structural parameters with electrochemical metrics to establish structure-property relationships. The CTS strategy offers a platform for next-generation electrode design, connecting rapid synthesis with high-performance energy storage.

## Figures and Tables

**Figure 1 micromachines-16-00490-f001:**
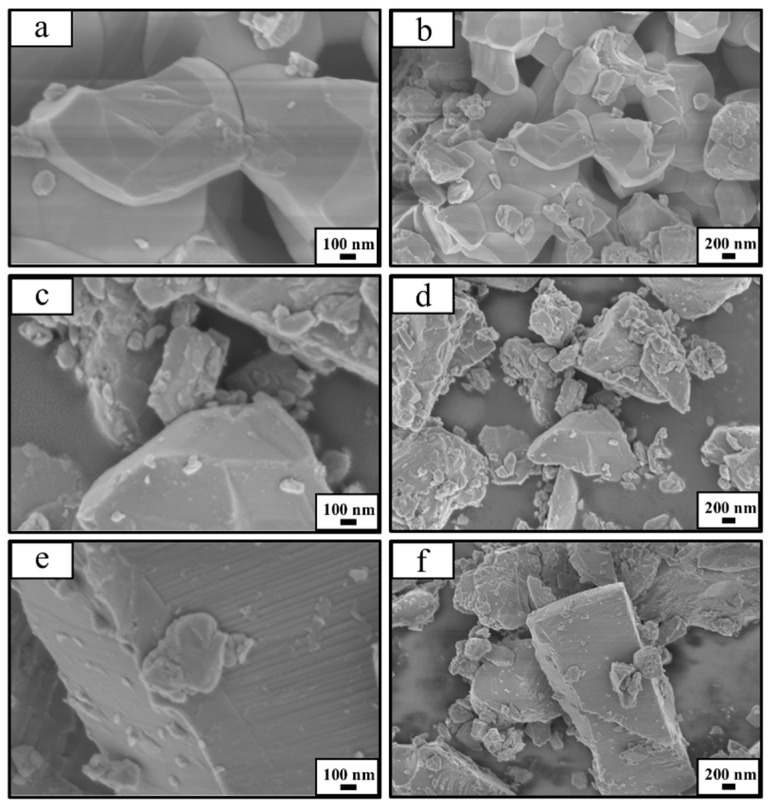
Structural and morphological characterization of the samples: scanning electron microscope (SEM) images of TNO-1100 (**a**,**b**), TNO-1200 (**c**,**d**), and TNO-1400 (**e**,**f**) at low and high magnifications, respectively.

**Figure 2 micromachines-16-00490-f002:**
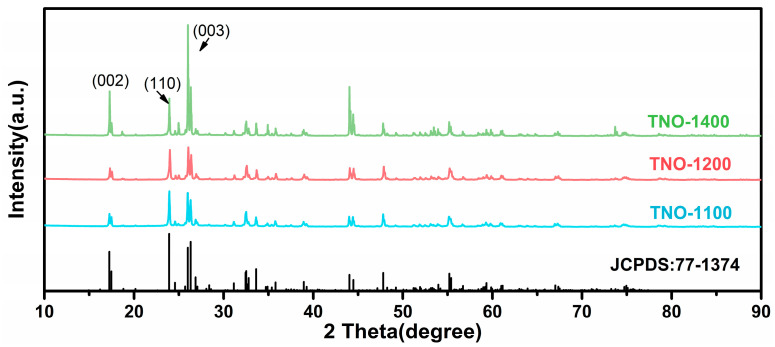
X-ray diffraction patterns of TNO-1100, TNO-1200, and TNO-1400 samples with reference diffraction lines (JCPDS 77-1374) for monoclinic TiNb_2_O_7_ (space group: C2/m). Characteristic peaks are indexed. Measurements were performed using Cu Kα radiation (λ = 0.154056 nm) with a scanning range of 2θ = 5–90° at 5°/min.

**Figure 3 micromachines-16-00490-f003:**
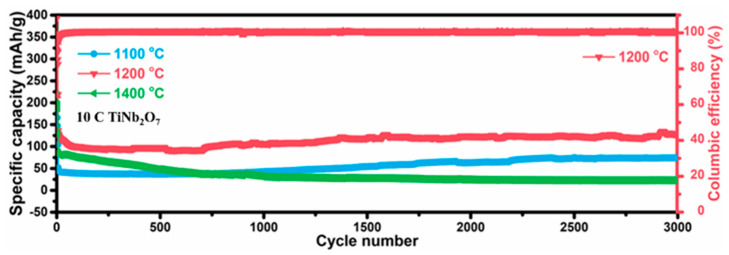
According to the results, the TNO-1200 electrode shows higher specific capacities than the TNO-1100 electrode and TNO-1400 electrode under the following conditions: current with 10 C; voltage window of 0.01–3.0 V after 3000 cycles.

**Figure 4 micromachines-16-00490-f004:**
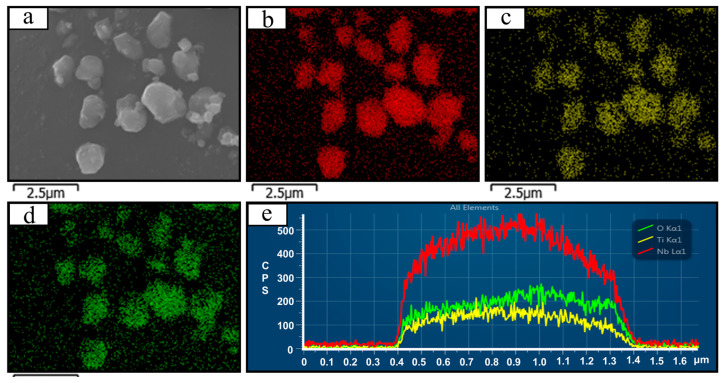
SEM images (**a**), SEM-EDS mapping (**b**–**d**), and line scan EDS (**e**) of the elemental distribution of TNO-1200.

**Figure 5 micromachines-16-00490-f005:**
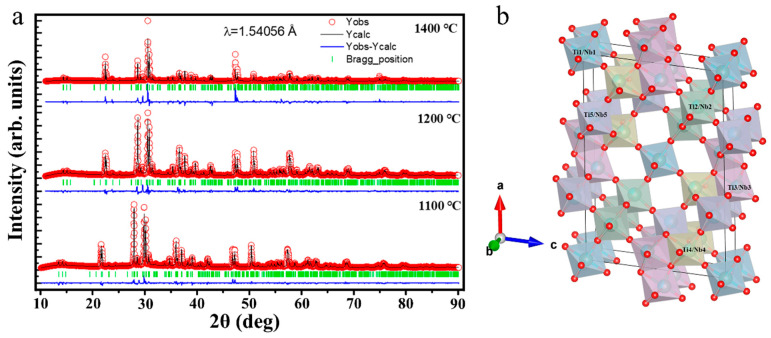
Rietveld refinement pattern of all samples generated using FullProf Software (Version 2.05), (**a**) TNO-1100, TNO-1200, and TNO-1400. (**b**) A reference crystal structure of sample TNO-1200 generated from refinement data.

**Table 1 micromachines-16-00490-t001:** Refined structural parameters (lattice constants a, b, and c, calculated unit cell volume V and atomic positions) by the Fullprof Suite from the XRD data measured at 300 K. Number in parenthesis is the estimated standard deviation of the last or the next last significant digit.

TiNb_2_O_7_ (Monoclinic, Space Group: C2/m)			
*T* (K)	300		
a, b, c (Å)	17.6869 (2)	3.8030 (0)	11.8976 (0)
α, β, γ (°)	90	95.32233 (3)	90
*V* (Å^3^)	796.830 (1)		
Atom	x	y	z
Nb1	0.0000 (0)	0.0000 (0)	0.0000 (0)
Ti1	0.0000 (0)	0.0000 (0)	0.0000 (0)
Nb2	0.18483 (36)	0.0000 (0)	0.18442 (58)
Ti2	0.18483 (36)	0.0000 (0)	0.18442 (58)
Nb3	0.07844 (33)	0.0000 (0)	−0.55762 (56)
Ti3	0.07844 (33)	0.0000 (0)	−0.55762 (56)
Nb4	0.89020 (36)	0.0000 (0)	0.25427 (59)
Ti4	0.89020 (36)	0.0000 (0)	0.25427 (59)
Nb5	0.29626 (43)	0.0000 (0)	−0.07822 (66)
Ti5	0.29626 (43)	0.0000 (0)	−0.07822 (66)
O1	0.18303 (0)	0.0000 (0)	−0.41559 (0)
O2	0.36568 (0)	0.0000 (0)	−0.25112 (0)
O3	0.59550 (0)	0.0000 (0)	−0.01283 (0)
O4	0.78760 (0)	0.0000 (0)	0.16717 (0)
O5	0.24841 (0)	0.0000 (0)	0.05577 (0)
O6	0.69940 (0)	0.0000 (0)	0.69177 (0)
O7	0.89633 (0)	0.0000 (0)	−0.08214 (0)
O8	0.01655 (0)	0.0000 (0)	−0.38232 (0)
O9	0.87235 (0)	0.0000 (0)	0.68015 (0)
O10	0.50000 (0)	0.0000 (0)	0.50000 (0)
O11	0.04006 (0)	0.0000 (0)	−0.13948 (0)
*R*_p_, *R*_wp_, *R*_exp_, χ^2^	5.08, 8.30, 3.68, 5.09		

**Table 2 micromachines-16-00490-t002:** Electrochemical performance comparison of TNO-1200 with other anode materials under fast-charging conditions.

Material	Current Density	Capacity (mAh/g)	Cycle Number	Retention Rate
TNO-1200 (This Work)	10 C	125	3000	98%
TiNb_2_O_7_ [11]	1 C	250	100	-
Ti_2_Nb_10_O_29_ [35]	10 C	100	500	95%
Oxygen-deficient TNO [26]	5 C	180	1000	85%

## Data Availability

The original contributions presented in this study are included in the article. Further inquiries can be directed to the corresponding author.

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
