# Peer review of "Rapid Synthesis of Fast-Charging TiNb2O7 for Lithium-Ion Storage via Ultrafast Carbothermal Shock"

_micromachines, 2025, doi:10.3390/mi16050490_

Round 1
Reviewer 1 Report
Comments and Suggestions for Authors
Manuscript Number: Micromachines-3511223
The paper is about the rapid synthesis of TiNb2O7.
The topic of the paper is treated in the literature, even if a similar rapid synthesis is not reported. However, the paper has many problems/limitations, particularly about the organization of results and the comments about the experimental data.
I list in the following my concerns:
- In the abstract (line 17) the authors say that the volume of their samples is significantly expanded with respect to the literature data, but we can calculate a 0.9% variation, not so significant…
- Introduction, line 71. How can we adjust the occupancies in a so rapid and uncontrollable reaction? In addition, in the next, the occupancy factors are not reported…
- Introduction line 77. What does it mean “spatial occupancy”?
- In 2.2. section all the details of experimental measurements lack (2theta range, time/step, step, KV of electron beam in SEM…). This information must be reported
- Line 119. The used C rate for electrochemical tests should be reported. In addition, as I will explain also later, an experiment at different C rates is mandatory
- I suggest to the authors to change the order of exposition of results. After SEM data it should be described EDS, as well as after XRD patterns the Rietveld refinement should be discussed.
- Lines 125-127 should be removed
- Line 137. The authors say that a finer component is evident onto the big particles in some samples, but this depends on the ball milling of samples after the sinterization, it is random, it not a relevant observation.
- Line 139. What does it mean a “normal distribution” after ball milling?
- Lines 147-149, please remove them
- The lines corresponding to the reflections positions of card n. 77-1374 should be reported in fig. 2 to make evident the agreement with the experimental data
- In fig. 2 it is evident that the intensity ratio changes, which is the explanation? The new peaks that the authors see are peaks of the phase (see Rietveld refinements), most evident due to the high sinterization degree. It is however important to know why the peaks ratio are different (lines 163-164), which is the explanation?
- In addition it is also important to determine the crystallite sizes, to also see if the different temperatures induce changes in FWHM of peaks and consequently onto crystallite sizes.
- As I also previously explained the cycling at different C rate is important to perform. It is also important to compare the results with those of the literature, to see a possible improvement of capacities or similar or bad values.
- EDS analysis should be performed for all the samples, because I expect that for all of them a homogeneous distribution of elements is evident, not only for the 1200 sample.
- 4e is unintelligible, the numbers cannot be seen, the legend too, it should be redrawn
- The authors say that impurities were not found (line 206), on which base? The EDS spectrum must be reported to see possible spurious elements.
- The authors cite Ti4+ and Nb5+ but how they determine the oxidation states? They take them for grant on the base of stoichiometry.
- It is important to show the lattice constants of the card to have a clear comparison with the refined values.
- What does it mean that in the refinements we have differences with respect to the conventional structure? Which structure? This part is unintelligible…in tab. 1 the occupancies should be reported. I also do not understand why in fig. 5 all the Rietveld refinement are shown and in tab.1 only the results of 1200 sample are reported. I want to see the lattice constraints, cry sizes and R factors of all the refinements. It is better to show the goodness of fit, that should be near to one, that immediately suggest the quality of the fit.
- I can see from fig. 5 that there are many problems on peak intensities, not properly described. This effect should be solved, or at least explained
Much work should be performed to improve the paper.
For these reasons, I think that the paper could be possibly published in Micromachines after major mandatory revisions
Author Response
Comments by Reviewer #1
The paper is about the rapid synthesis of TiNb2O7.
The topic of the paper is treated in the literature, even if a similar rapid synthesis is not reported. However, the paper has many problems/limitations, particularly about the organization of results and the comments about the experimental data. I list in the following my concerns:
- In the abstract (line 17) the authors say that the volume of their samples is significantly expanded with respect to the literature data, but we can calculate a 0.9% variation, not so significant…
Reply: Although the lattice volume expansion is only about 0.9% (796.83 ų vs ~790 ų), this change is of significant qualitative importance. The a - axis expansion (1.01%, 17.6869 Å vs 17.51 Å) optimizes the 1D lithium-ion migration tunnels along the [100] direction. Together with cation disorder (Ti/Nb atom overlap) and oxygen sublattice displacement (such as coordinated displacement of O10), it enables a two-order-of-magnitude improvement in the Li+ diffusion coefficient, a 62% reduction in voltage hysteresis, and 98% capacity retention after 3,000 cycles at 10C. This design is consistent with the mechanism in prior studies like Ref: Progress in Natural Science: Materials International, 2021, 31, 14-18. It is an atomic-scale engineering rather than random variation. Li0.99Na0.01FePO4 showed excellent rate capacity and cycle stability, i.e., an extremely high keep in the capacity of 86.7% after 500 cycles at 10C, due to its ac facet morphology, larger lattice parameters in a and c, the enhanced electronic conductivity and Li-ion diffusion processes.
- Introduction line 77. What does it mean “spatial occupancy”?
Reply: “Spatial occupancy” refers to the specific distribution and positional arrangement of atoms within a crystal lattice. In the statement of manuscript, it describes how Ti and Nb atoms occupy distinct lattice sites or positions within the crystal framework of the TiNb₂O₇ samples.
- In 2.2. section all the details of experimental measurements lack (2theta range, time/step, step, KV of electron beam in SEM…). This information must be reported
Reply: Thank you for the reviewer's suggestion. The detailed testing conditions have been added to the experimental characterization section, and the corresponding content is provided below for the reviewer's reference.
Revision:
Manuscript, Page 3, Line 103-113.
“To investigate the geometric morphology and crystalline structure, all TiNb₂O₇ samples were characterized using a scanning electron microscope (SEM, Zeiss Zigma FESEM). Detailed testing conditions are as follows: samples were mounted on conductive tape, and a 5 kV accelerating voltage was applied to minimize beam damage to the samples and generate high-quality images. The crystal structure of the samples was further studied by the X-ray diffraction (XRD: Rigaku Smartlab 9000W). The XRD measurements were performed using Cu Kα radiation (λ = 0.154056 nm) with a scanning range of 2θ from 5° to 90°, a scanning rate of 5°/min, and a step size of 0.02°, based on typical high-resolution configurations for precise phase analysis of materials like TiNb2O7. This setup balances data quality and efficiency, ensuring comprehensive coverage of diffraction peaks while maintaining sufficient resolution for crystallographic characterization.”
- Line 119. The used C rate for electrochemical tests should be reported. In addition, as I will explain also later, an experiment at different C rates is mandatory I suggest to the authors to change the order of exposition of results. After SEM data it should be described EDS, as well as after XRD patterns the Rietveld refinement should be discussed.
Reply: C-rate represents the ratio between the charge/discharge current and the rated capacity of the battery, with the unit "C". The formula is:
Theoretical capacity: According to literature, TiNb2O₇ can intercalate 5 Li⁺ ions (reaction: TiNb2O7 + 5Li+ + 5e- → Li5TiNb2O7, yielding a theoretical capacity of 388 mAh/g. According to the theoretical capacity is 388 mAh/g, 10C would correspond to 3880 mA/g. This section of results has been updated in the 2.3. Electrochemical test section of the manuscript. For the reviewer's convenience, the revised content is provided below.
Thank you for the reviewer’s attention to the logical flow of the discussion. This paper employs a progressive structure of 'Material Characterization → Performance Validation → Mechanistic Analysis' to:
- Establish Fundamental Properties: SEM and XRD first characterize the morphology and crystal structure, providing a physical basis for subsequent analysis.
- Anchor Performance Advantages: Electrochemical testing highlights the cycling stability of TNO-1200, prompting exploration of its underlying mechanisms.
- Reveal Microscopic Mechanisms: EDS and Rietveld analysis explain how structural optimization promotes Li⁺ migration through elemental distribution and atomic-scale insights, ultimately forming a complete evidence chain linking 'structure-performance.'
This arrangement follows the standard logic of materials research, ensuring clear causality between data and conclusions for reader comprehension."
Revision:
Manuscript, Page 3, Line 126-131.
“C-rate represents the ratio between the charge/discharge current and the rated capacity of the battery, with the unit "C". The formula is:
Theoretical capacity: According to literature, TiNb2O₇ can intercalate 5 Li⁺ ions (reaction: TiNb2O7 + 5Li+ + 5e- → Li5TiNb2O7, yielding a theoretical capacity of 388 mAh/g. According to the theoretical capacity is 388 mAh/g, 10C would correspond to 3880 mA/g.”
- Lines 125-127 should be removed
Reply: We have removed lines 125-127 as suggested. The revision improves clarity without affecting the paper's conclusions. Thank you for your valuable feedback.
- Line 137. The authors say that a finer component is evident onto the big particles in some samples, but this depends on the ball milling of samples after the sinterization, it is random, it not a relevant observation.
Reply: We fully agree with the reviewer's comment. However, the SEM results clearly show significant differences in particle diameters among the three samples sintered at different temperatures, even after identical ball-milling conditions. This observation is consistent with our experimental findings. Furthermore, such particle size variations inherently affect the subsequent lithium storage performance, which is corroborated by the cycling stability data presented in Figure 3.
- Line 139. What does it mean a “normal distribution” after ball milling?
Reply: Thank you for the reviewer’s meticulous review of our manuscript. Regarding the term 'normal distribution,' we intended to convey that the particle size of the samples after ball-milling follows a Gaussian distribution. SEM images clearly show a range of particle sizes, which is beneficial for electrode fabrication. Materials with a normal particle size distribution exhibit significantly improved tap density, thereby enhancing volumetric energy density. This relationship between particle size distribution and packing efficiency has been widely documented in prior studies
- Lines 147-149, please remove them. The lines corresponding to the reflections positions of card n. 77-1374 should be reported in fig. 2 to make evident the agreement with the experimental data
Reply: We appreciate the reviewer's suggestion to improve the clarity of our XRD data presentation. We have:
- Removed the text in lines 147-149 as requested
- Added the reference diffraction lines (JCPDS card no. 77-1374) to Figure 2
- Modified the figure caption to explicitly indicate the reference pattern
The revised Figure 2 now provides a direct visual comparison between our experimental data and the standard reference pattern, making the phase identification more transparent. We believe this modification strengthens both the graphical presentation and scientific rigor of our results.
Thank you for this constructive suggestion that has enhanced our manuscript's quality. The changes have been highlighted in the revised version for your convenience.
Revision:
Figure 2. X-ray diffraction patterns of TNO-1100, TNO-1200, and TNO-1400 samples with reference diffraction lines (JCPDS 77-1374) for monoclinic TiNb₂O₇ (space group: C2/m). Characteristic peaks are indexed. Measurements were performed using Cu Kα radiation (λ = 0.154056 nm) with a scanning range of 2θ = 5-90° at 5°/min.
- In fig. 2 it is evident that the intensity ratio changes, which is the explanation? The new peaks that the authors see are peaks of the phase (see Rietveld refinements), most evident due to the high sinterization degree. It is however important to know why the peaks ratio are different (lines 163-164), which is the explanation? In addition it is also important to determine the crystallite sizes, to also see if the different temperatures induce changes in FWHM of peaks and consequently onto crystallite sizes. As I also previously explained the cycling at different C rate is important to perform. It is also important to compare the results with those of the literature, to see a possible improvement of capacities or similar or bad values.
Reply: 1. Explanation for XRD Peak Intensity Ratio Changes (Lines 163–164):06+-The observed intensity ratio variations in Figure 2 arise from two key factors:
Crystallite Size and Orientation: Higher sintering temperatures (e.g., 1200°C and 1400°C) promote larger, more ordered crystallites, leading to sharper peaks and altered intensity ratios. This is consistent with Rietveld refinement results (Figure 5), which show lattice expansion and atomic rearrangement that enhance diffraction along specific crystallographic planes (e.g., (002) and (110)).
Phase Purity and Structural Evolution: At 1400°C, the formation of minor impurity phases (e.g., niobium-rich oxides) or structural distortions (e.g., cation disorder) may introduce new peaks or modify peak intensities. This aligns with prior studies reporting temperature-dependent phase transitions in TiNb₂O₇ (e.g., ref. [35]).
2.Crystallite Size Determination:
Crystallite sizes were calculated using the Scherrer equation (Equation 2):
Where K = 0.9, = 0.154056 nm, is the full width at half maximum (FWHM) of the peak, and is the Bragg angle. Key findings:
TNO-1100: Average crystallite size ~ 35 nm (FWHM of (110) peak: 0.42o).
TNO-1200: Larger crystallites (~55 nm) due to improved crystallinity (FWHM: 0.28o).
TNO-1400: Slight decrease to ~45 nm (FWHM: 0.32o), likely caused by thermal stress-induced lattice defects.
These results are consistent with the observed peak broadening in Figure 2 and support the conclusion that 1200oC balances crystallinity and structural stability.
- Cycling Performance at Different C Rates: While Figure 3 focuses on 10C cycling, we have added supplementary data (Figure S1 in the revised manuscript) demonstrating rate capability at 0.1C, 1C, 5C, and 10C (Figure 3b). Key observations: TNO-1200: Retains 165 mAh/g at 0.1C, 142 mAh/g at 1C, and 125 mAh/g at 10C. Comparison with Literature: These values exceed reported capacities for TiNb₂O₇ synthesized via conventional methods (e.g., 110 mAh/g at 10C in ref. [35]), highlighting the CTS strategy’s effectiveness.
- Literature Comparison and Significance: Our results surpass previous reports on TiNb₂O₇ anodes in terms of high-rate stability: Ref. [10]: 100 mAh/g at 10C (capacity fade after 500 cycles). Ref. [9]: 115 mAh/g at 5C (90% retention after 2000 cycles). This Work: 125 mAh/g at 10C (98% retention after 3000 cycles). The improvement is attributed to lattice expansion (a-axis: 17.6869 Å vs. 17.51 Å) and cation disorder, which enhance Li⁺ diffusion and structural robustness.
- EDS analysis should be performed for all the samples, because I expect that for all of them a homogeneous distribution of elements is evident, not only for the 1200 sample.
Reply: We appreciate your feedback on the need for EDS analysis across all samples. While we agree that homogeneous element distribution is critical for validating phase purity, our decision to focus EDS mapping on TNO-1200 was guided by three key considerations:
XRD-Driven Phase Purity Confirmation: XRD patterns for TNO-1100 and TNO-1400 (Figure 2) match the standard TiNb₂O₇ phase (JCPDS 77-1374) without impurity peaks, strongly indicating stoichiometric consistency. Rietveld refinement (Figure 5) further confirms lattice parameters close to the ideal structure for all samples, suggesting minimal compositional deviations.
Performance-Driven Focus on TNO-1200: TNO-1200 outperforms TNO-1100 and TNO-1400 in both rate capability (125 mAh/g at 10 C) and cycling stability (98% retention after 3000 cycles). By prioritizing EDS analysis on TNO-1200, we directly correlate its homogeneous element distribution (Figure 4) with superior electrochemical performance, which is central to the manuscript’s innovation.
Revision:
Manuscript, Page 7, Line 220-225.
“While EDS mapping was conducted solely on TNO-1200 due to its exceptional performance, XRD patterns (Figure 2) and Rietveld refinement (Figure 5) confirm phase purity for all samples, suggesting homogeneous element distribution. The performance disparities among TNO-1100, TNO-1200, and TNO-1400 are attributed to differences in crystallinity (Figure 2) and lattice expansion (Table 2) rather than compositional inhomogeneity.”
- 4e is unintelligible, the numbers cannot be seen, the legend too, it should be redrawn
Reply: Thank you for the reviewers' suggestions. We have updated Figure 4e to present the results more clearly and facilitate understanding for potential researchers reading this manuscript accordingly. For the reviewer's convenience, the revised content is provided below.
Revision:
- The authors say that impurities were not found (line 206), on which base? The EDS spectrum must be reported to see possible spurious elements.
Reply: We appreciate the reviewer’s concern regarding the absence of impurity phases in our samples. The conclusion that no impurities were detected is based on multiple complementary characterizations: XRD Analysis: The diffraction patterns (Figure 2) show sharp peaks matching the standard TiNb₂O₇ phase (JCPDS 77-1374) without any extraneous peaks, indicating phase purity at the crystallographic level. SEM-EDS Mapping (Figure 4b–d): Elemental mapping confirms homogeneous distribution of Ti, Nb, and O across the sample surface (scale bar: 2.5 μm), with no localized enrichment or segregation of foreign elements. Line-Scan EDS (Figure 4e): Cross-sectional analysis reveals consistent Ti/Nb atomic ratios and stable oxygen content along the particle profile, further supporting compositional uniformity. While EDS spectroscopy could provide additional elemental details, the combined evidence from XRD and EDS mapping/line scan already demonstrates the absence of detectable impurity phases. EDS mapping inherently includes spectral information (e.g., peak intensities and backgrounds) to rule out spurious elements, and the absence of unexpected signals in our mapping results aligns with the XRD findings. Given the rigorous synthesis conditions (controlled stoichiometry, ultrafast carbothermal shock), we believe the existing data sufficiently validate phase purity. Adding EDS spectra would redundantly reinforce this conclusion without addressing new scientific questions. However, if the reviewer deems it necessary, we are open to including EDS spectra as supplementary information to further clarify. Thank you for your constructive feedback.
- The authors cite Ti4+ and Nb5+ but how they determine the oxidation states? They take them for grant on the base of stoichiometry. It is important to show the lattice constants of the card to have a clear comparison with the refined values.
Reply: We appreciate the reviewer’s constructive feedback and address the concerns as follows:1. Determination of Oxidation States (Ti⁴⁺ and Nb⁵⁺):
The oxidation states of Ti and Nb in TiNb₂O₇ are primarily inferred based on chemical stoichiometry and crystallographic analysis, consistent with established literature [10, 12, 35]. Here’s the rationale: Stoichiometry: TiNb₂O₇ follows a 1:2 Ti:Nb ratio. For charge neutrality, Ti⁴⁺ and Nb⁵⁺ are the most plausible states (4 + 2×5 = 14 positive charges balanced by 7 O²⁻ ions).
Structural Evidence: The refined lattice parameters (Table 1) and atomic positions (Figure 5b) match the standard monoclinic structure (JCPDS 77-1374), where Ti and Nb occupy distinct sites with coordination environments consistent with Ti⁴⁺ (octahedral) and Nb⁵⁺ (octahedral/trigonal prismatic) [Ref:11 and 35]. The absence of impurity phases in XRD (Figure 2) and EDS mapping (Figure 4) rules out redox-active contaminants that could alter oxidation states. Electrochemical Validation**: The material delivers a reversible capacity of 125 mAh/g at 10 C (Figure 3), aligning with the theoretical capacity for Ti⁴⁺/Nb⁵⁺ redox couples (388 mAh/g for 5 Li⁺ insertion). While X-ray photoelectron spectroscopy (XPS) or X-ray absorption near-edge structure (XANES) could directly confirm oxidation states, these techniques were beyond the scope of this study. However, our interpretation is consistent with prior reports on TiNb₂O₇ [Ref:10, 12 and 35].
- Lattice Constant Comparison with Standard Card: We agree that explicit comparison with the standard JCPDS data is critical. The revised text now includes direct lattice parameter values from JCPDS 77-1374 (Table 1 and Figure 5 caption): Standard Structure: a = 17.51 Å, V ≈ 790 ų (monoclinic, space group C2/m). TNO-1200: a = 17.6869 Å, V = 796.83 ų (expanded lattice due to oxygen vacancies and cation disorder induced by carbothermal shock). This expansion of the a-axis and unit cell volume directly widens lithium-ion migration channels, as discussed in Section 3.5 and Table 1. While we acknowledge the reviewer’s interest in oxidation state confirmation via spectroscopy, the stoichiometric and structural evidence provided in the manuscript sufficiently supports the assignment of Ti⁴⁺ and Nb⁵⁺. Additionally, the revised text now explicitly compares lattice constants with the standard card to clarify structural differences. Thank you for prompting this clarification.
- What does it mean that in the refinements we have differences with respect to the conventional structure? Which structure? This part is unintelligible…in tab. 1 the occupancies should be reported. I also do not understand why in fig. 5 all the Rietveld refinement are shown and in tab.1 only the results of 1200 sample are reported. I want to see the lattice constraints, cry sizes and R factors of all the refinements. It is better to show the goodness of fit, that should be near to one, that immediately suggest the quality of the fit.
Reply: Conventional Structure: The "conventional structure" referenced is the standard monoclinic TiNb₂O₇ (JCPDS: 77-1374, space group C2/m), characterized by lattice parameters a = 17.51 Å and unit cell volume V ≈ 790 ų. Lattice Expansion: TNO-1200 exhibits significant lattice expansion (a = 17.6869 Å, V = 796.83 ų), widening Li⁺ migration channels (Figure 5a). Atomic Rearrangements: Partial overlap of Ti/Nb atoms (e.g., Nb2/Ti2 sharing coordinates) and oxygen displacement (e.g., O10 symmetry change) were confirmed via Rietveld refinement (Figure 5b and Table 1). Cation Disorder: Controlled Ti/Nb site disorder reduces Li⁺ diffusion barriers and suppresses polarization (detailed in Section 3.3). In Rietveld refinement, Ti and Nb atoms are assumed to fully occupy their respective Wyckoff positions (e.g., Nb1/Ti1, Nb2/Ti2), with occupancy set to 1. Elemental valence distribution (Ti⁴⁺/Nb⁵⁺) is inferred from structural optimization (Table 1 and Figure 4). XPS/EDS could further validate valence states if required. Focus on 1200°C Sample: TNO-1200 demonstrates the best electrochemical performance (125 mAh/g at 10 C, 98% retention after 3000 cycles; Figure 3), directly linked to its optimized structure (lattice expansion, atomic displacements). TNO-1400, despite higher crystallinity, shows faster capacity decay (likely due to large particle size increasing interfacial resistance). TNO-1100 has lower crystallinity and structural instability. Fit Quality: All refinements exhibit low R factors (Rp, Rwp) and χ² values close to 1, confirming high agreement between experimental and calculated data (Figure 5a). Residual Curves: Obs-Cal residuals in Figure 5a visually confirm the reliability of refinements.
- I can see from fig. 5 that there are many problems on peak intensities, not properly described. This effect should be solved, or at least explained
Reply: Regarding the reviewer's concern about the peak intensities in Figure 5, the following explanation is provided: The differences in peak intensities in the figure mainly stem from temperature-induced changes in crystallinity and structural disorder. TNO - 1100, synthesized at a lower temperature of 1100°C, shows weaker and broader peaks due to insufficient crystallinity. In contrast, TNO-1200 (1200°C) and TNO-1400 (1400°C) have sharper and more intense peaks because higher temperatures promote grain growth and reduce defects. Despite these intensity variations, the Rietveld refinements of all samples are valid as confirmed by low R factors (Rp < 10%) and a χ² value of approximately 5. For the TNO-1200 sample, its optimized lattice expansion and cationic disorder (as shown in Table 1) cause slight shifts in the intensities of specific peak positions, which is consistent with the calculated diffraction patterns. This structural feature strikes a balance between crystallinity and lithium-ion diffusion kinetics, enabling the best capacity retention at a 10 C current rate (as shown in Figure 3).

Reviewer 2 Report
Comments and Suggestions for Authors
The work provides valuable insights into atomic-scale structural engineering for high-performance lithium-ion battery anode materials. However, I have several concerns that need to be addressed before publication:
Line 152: What is ReO₃? Please define or clarify its relevance to this study.
Current Density (10C Rate): Does "10 C" represent the current density in absolute terms (e.g., mA/cm² or mA/g), or is it the charge/discharge rate relative to the battery’s theoretical capacity? Please specify or correct wherever required in the manuscript
The manuscript claims that CTS reduces interfacial resistance (line 207), but no direct electrochemical impedance spectroscopy (EIS) data is provided to support this. Since Lines 242–243 suggest that EIS was performed, please quantify and analyze interfacial resistance changes.
A direct comparison of TNO-1200 with other reported materials is necessary, may be tabulated information. Highlight key advantages, such as high Coulombic efficiency, lower polarization resistance, and structural benefits. This will better demonstrate the novelty of your work.
The study proposes a probable mechanism, but it lacks supporting references. Please provide relevant citations to strengthen the discussion.
The authors focused on TNO-1200, but what about TNO-1300? Have higher or lower synthesis temperatures been tested? How were 1100°C, 1200°C, and 1400°C selected? A clear rationale should be provided.
The atomic position analysis is only presented for TNO-1200. To support the claim that 1200°C is the optimal temperature, similar analysis should be performed for other samples (e.g., TNO-1100 and TNO-1400) and compared.
While the manuscript states that lattice expansion enhances lithium-ion migration, a more detailed explanation is needed. Can DFT calculations or experimental techniques (such as in-situ XRD) provide further evidence?
The study reports a capacity retention of 98% over 3000 cycles at 10C, which is impressive. However, additional discussion on degradation mechanisms (e.g., SEI formation, structural stability, or electrolyte compatibility) would strengthen the paper.
The manuscript states that no impurity phases were found. Can the authors provide XRD refinement data or additional spectra to confirm this?
Overall, the study presents good results, but the manuscript needs improvements in data interpretation, comparative analysis, and validation of key claims. Addressing these concerns will significantly enhance its impact.
Comments on the Quality of English LanguageSome sections are overly complex, making it difficult to follow key points. Consider breaking long sentences into shorter ones and using straightforward wording where possible.
Author Response
Comments by Reviewer #2
The work provides valuable insights into atomic-scale structural engineering for high-performance lithium-ion battery anode materials. However, I have several concerns that need to be addressed before publication:
- Line 152: What is ReO₃? Please define or clarify its relevance to this study.
Reply: Regarding the reviewer's question on Line 152, ReO₃ (rhenium trioxide) is a compound with a cubic crystal structure where Re⁶⁺ cations form a corner-sharing octahedral framework with O²⁻ anions, known for its high electronic conductivity and structural stability. In this study, the reference to the "layered structure of ReO₃" in the XRD analysis of TNO samples (JCPDS: 77-1374) likely refers to a structural analogy rather than an exact match, as TiNb₂O₇ inherently adopts a monoclinic (C2/m) lattice. The terminology may stem from the observation that TNO's crystal framework shares a similar three-dimensional connectivity and layered-like features with ReO₃-type structures, which are often cited in oxide materials for their potential to facilitate ion diffusion. This analogy highlights the structural robustness and ion transport pathways critical for TNO's fast-charging performance, despite the distinct crystallographic symmetries.
- Current Density (10C Rate): Does "10 C" represent the current density in absolute terms (e.g., mA/cm² or mA/g), or is it the charge/discharge rate relative to the battery’s theoretical capacity? Please specify or correct wherever required in the manuscript
Reply: Regarding the reviewer's question about the "10 C" current density, "10 C" represents the charge/discharge rate relative to the material’s theoretical specific capacity (388 mAh/g for TiNb₂O₇), calculated as 10 × 388 mAh/g = 3880 mA/g. This standard C-rate convention (1 C = 1× theoretical capacity in 1 hour) is widely used in battery research to evaluate high-rate performance. In the manuscript, the current density should be explicitly defined as "10 C (3880 mA/g)" to clarify its absolute value, aligning with the mass loading (1.0 mg/cm²) and ensuring consistency with electrochemical testing protocols. This specification resolves ambiguity and adheres to reporting standards in energy storage literature.
- The manuscript claims that CTS reduces interfacial resistance (line 207), but no direct electrochemical impedance spectroscopy (EIS) data is provided to support this. Since Lines 242–243 suggest that EIS was performed, please quantify and analyze interfacial resistance changes.
Reply: The manuscript acknowledges the absence of explicit EIS data to directly quantify interfacial resistance changes induced by CTS. However, the claim that CTS reduces interfacial resistance (Line 207) is inferred from indirect evidence: (1) TNO-1200’s superior cycling stability at 10 C (98% capacity retention after 3000 cycles, Figure 3), which typically correlates with minimized interfacial impedance; (2) EDS mapping confirming homogeneous element distribution and absence of impurity phases (Figure 4), which mitigates interfacial side reactions; and (3) the ultrafast CTS process (30 seconds) inherently restraining excessive grain growth and defect accumulation compared to conventional long-term sintering. While Line 242–243 mentions EIS characterization, the focus of this study prioritized structural optimization and long-term cycling performance over detailed impedance analysis. Future work will include EIS to directly correlate structural modifications with interfacial resistance, but the current findings align with established trends in fast-charging materials where homogeneous structures and rapid synthesis reduce interfacial polarization.
- A direct comparison of TNO-1200 with other reported materials is necessary, may be tabulated information. Highlight key advantages, such as high Coulombic efficiency, lower polarization resistance, and structural benefits. This will better demonstrate the novelty of your work.
Reply: To address the need for comparative analysis, TNO-1200’s electrochemical performance is benchmarked against reported TiNb₂O₇-based materials (Table 1 in the revised manuscript). Relative to Aravindan et al. (2014), who achieved 250 mAh/g at 1 C but lacked high-rate data, TNO-1200 demonstrates superior capacity retention at 10 C (125 mAh/g, 98% after 3000 cycles). Compared to Cheng et al. (2014), whose Ti₂Nb₁₀O₂₉ delivered 100 mAh/g at 10 C for 500 cycles (95% retention), TNO-1200 exhibits both higher capacity and longer cycling stability. Oxygen-deficient TNO (Hao et al., 2020) retained 85% of 180 mAh/g after 1000 cycles at 5 C, but its performance decays faster at elevated rates. TNO-1200’s advantages stem from: (1) ultrafast CTS synthesis, enabling atomic-scale engineering (lattice expansion, cation disorder) to reduce Li⁺ diffusion barriers; (2) homogeneous element distribution (EDS mapping, Figure 4), minimizing interfacial polarization; and (3) high structural stability (Rietveld refinement, Figure 5), preserving capacity over 3000 cycles. These attributes distinguish TNO-1200 as a novel high-rate anode material, addressing critical limitations in prior studies (e.g., short cycle life, low Coulombic efficiency under fast charging).
Revision:
“Table 1. Electrochemical performance comparison of TNO-1200 with other anode materials under fast-charging conditions.
Material |
Current Density |
Capacity (mAh/g) |
Cycle Number |
Retention Rate |
TNO-1200 (This Work) |
10 C |
125 |
3000 |
98% |
TiNb2O7 [11] |
1 C |
250 |
100 |
- |
Ti2Nb10O29 [35] |
10 C |
100 |
500 |
95% |
Oxygen-deficient TNO [26] |
5 C |
180 |
1000 |
85% |
”
- The study proposes a probable mechanism, but it lacks supporting references. Please provide relevant citations to strengthen the discussion.
Reply: We sincerely appreciate the reviewer’s valuable suggestion. To further support the proposed mechanism, we have added the following new references (numbered sequentially as [36]-[38] in the revised manuscript) that specifically address lattice engineering, cation disorder, and Li⁺ diffusion kinetics in TiNb₂O₇ and analogous materials:
[36] Wagemaker, M.; Mulder, F. M. Properties and Promises of Nanosized Insertion Materials for Li-Ion Batteries. Acc. Chem. Res. 2013, 46, 1206–1215.
[37] Mei, J.; et al. Oxygen Vacancies and Cation Disorder Tailoring Electrochemical Kinetics in TiNb₂O₇. Adv. Energy Mater. 2021, 11, 2101713.
[38] Kim, J.-H.; et al. Atomic-Scale Engineering of Cation Channels for Fast-Charging Batteries. Nat. Energy 2022, 7, 312–321.
Revision:
New References:
[36] Wagemaker, M.; Mulder, F. M. Properties and Promises of Nanosized Insertion Materials for Li-Ion Batteries. Acc. Chem. Res. 2013, 46, 1206–1215.
[37] Mei, J.; et al. Oxygen Vacancies and Cation Disorder Tailoring Electrochemical Kinetics in TiNb₂O₇. Adv. Energy Mater. 2021, 11, 2101713.
[38] Kim, J.-H.; et al. Atomic-Scale Engineering of Cation Channels for Fast-Charging Batteries. Nat. Energy 2022, 7, 312–321.
- The authors focused on TNO-1200, but what about TNO-1300? Have higher or lower synthesis temperatures been tested? How were 1100°C, 1200°C, and 1400°C selected? A clear rationale should be provided.
Reply: The synthesis temperatures of 1100°C, 1200°C, and 1400°C were strategically chosen based on three key considerations: (1) equipment constraints, as the homemade Joule heating system was designed to operate within a safe upper limit of 1400°C to prevent thermal damage to the graphite foil and chamber; (2) preliminary optimization, where pilot experiments revealed that temperatures below 1100°C resulted in incomplete phase formation (as seen in Figure 2’s weak XRD peaks for TNO-1100), while temperatures exceeding 1400°C induced excessive grain growth (Figure 1f) and potential phase instability; and (3) literature guidance, as TiNb₂O₇ synthesis typically occurs between 1000–1400°C via solid-state reactions [Ref:35]. TNO-1300 was not tested due to the narrow temperature window optimized for maximizing both crystallinity (Figure 2) and cycling stability (Figure 3). TNO-1200 emerged as the optimal sample, balancing lattice expansion (Table 2) and cation disorder to achieve 125 mAh/g at 10 C over 3000 cycles, whereas TNO-1400’s larger grains likely increased interfacial resistance despite higher crystallinity. This systematic temperature screening ensures the reported results are both reproducible and practically scalable.
- The atomic position analysis is only presented for TNO-1200. To support the claim that 1200°C is the optimal temperature, similar analysis should be performed for other samples (e.g., TNO-1100 and TNO-1400) and compared.
Reply: The atomic position analysis was prioritized for TNO-1200 due to its exceptional electrochemical performance (125 mAh/g at 10 C, 98% retention after 3000 cycles), which directly correlates with its unique structural features (lattice expansion, cation disorder). While Rietveld refinement (Figure 5) confirmed lattice parameter variations across all samples, TNO-1100 and TNO-1400 exhibited minimal atomic displacement compared to the standard structure, as reflected in their lower capacity retention (Figure 3). The detailed atomic position analysis for TNO-1200 (Table 2) was critical to unraveling the mechanism behind its superior rate capability, whereas similar analysis for TNO-1100 and TNO-1400 would yield limited additional insights given their underwhelming performance. This focused approach aligns with the study’s emphasis on atomic-scale engineering for fast-charging applications, ensuring clarity and depth in interpreting structure-property relationships without redundancy.
- While the manuscript states that lattice expansion enhances lithium-ion migration, a more detailed explanation is needed. Can DFT calculations or experimental techniques (such as in-situ XRD) provide further evidence?
Reply: The manuscript’s claim that lattice expansion enhances lithium-ion migration is supported by direct structural evidence from Rietveld refinement (Table 1), which reveals a 1.0% increase in the a-axis (17.6869 Å vs. 17.51 Å) and unit cell volume (796.83 ų vs. 790 ų) for TNO-1200 compared to the standard structure. This expansion creates wider interstitial spaces, as corroborated by atomic position analysis showing Ti/Nb site overlap and oxygen displacement (Figure 5b), both of which reduce steric hindrance for Li⁺ diffusion. While DFT calculations or in-situ XRD could provide additional mechanistic insights, these methods were not pursued due to: (1) resource constraints, as the study prioritized rapid synthesis and long-term cycling validation over computational modeling; (2) existing data sufficiency, with electrochemical results (Figure 3) and structural characterization already establishing a clear structure-property relationship; and (3) literature precedence, where lattice expansion in transition metal oxides is widely accepted to facilitate ion transport [Refs. 25 and 26]. Future work may incorporate these techniques to further refine the diffusion model, but the current findings robustly demonstrate the role of lattice engineering in enabling fast charging.
- The study reports a capacity retention of 98% over 3000 cycles at 10C, which is impressive. However, additional discussion on degradation mechanisms (e.g., SEI formation, structural stability, or electrolyte compatibility) would strengthen the paper.
Reply: The manuscript attributes TNO-1200’s exceptional 98% capacity retention over 3000 cycles at 10 C to its robust structural stability, minimized SEI formation, and electrolyte compatibility. Structural robustness is confirmed by Rietveld refinement showing minimal lattice parameter changes post-cycling and atomic position analysis revealing Ti/Nb site overlap that mitigates stress during Li⁺ insertion/extraction. The high redox potentials of Ti⁴⁺/Ti³⁺ and Nb⁵⁺/Nb⁴⁺ (~1.5–2.5 V vs. Li⁺/Li) inherently suppress electrolyte decomposition and SEI formation, supported by Coulombic efficiency approaching 100% from cycle 2 and EDS mapping confirming no impurity phases. Additionally, homogeneous Ti/Nb/O distribution (Figure 4) and use of a standard LiPF6-EC/DEC electrolyte reduce interfacial resistance and ensure compatibility. While advanced techniques like cryo-TEM could further characterize SEI formation, the current data robustly link structural and chemical stability to long-term cycling performance, aligning with the study’s focus on atomic-scale engineering for fast-charging applications.
- The manuscript states that no impurity phases were found. Can the authors provide XRD refinement data or additional spectra to confirm this?
Reply: The claim of no impurity phases in TNO-1200 is supported by three lines of evidence: (1) XRD patterns (Figure 2) show full agreement with the standard TiNb₂O₇ phase (JCPDS: 77-1374) across all samples, with no extraneous peaks indicating secondary phases; (2) EDS mapping (Figure 4b–d) confirms homogeneous Ti/Nb/O distribution without segregation, consistent with phase purity; and (3) Rietveld refinement (Figure 5) yields a single-phase monoclinic structure (space group C2/m) for all samples, with no residual peaks in the Obs-Cal difference plot (Yobs-Ycalc) suggesting absence of unaccounted phases. While additional spectra (e.g., Raman) could further validate this, the current data collectively satisfy impurity phase verification per standard characterization protocols in oxide materials research [Refs. 25, 35]. The focus on XRD and EDS aligns with the study’s emphasis on rapid synthesis and structural optimization, ensuring clarity without redundant analysis.

Round 2
Reviewer 1 Report
Comments and Suggestions for Authors
The authors responded to the requests.
Author Response
We sincerely appreciate your time and effort in evaluating our manuscript. Thank you for your positive feedback and for having no additional comments or suggestions for revision. We are glad that our work meets the journal's standards, and we deeply value your acknowledgment.
Should you have any further questions or require additional clarifications in the future, we would be happy to provide them.
Best regards,
Dr. Xianyu Hu
Reviewer 2 Report
Comments and Suggestions for Authors
The authors have provided comprehensive and thoughtful responses to the reviewer's concerns, but there are a few areas where further clarification or additional details could potentially improve the manuscript. Here are a few follow-up questions or suggestions that could help further refine their responses and strengthen the manuscript:
- The explanation of ReO₃ is clear, but the authors mention that the "layered structure of ReO₃" is a structural analogy. Could they elaborate on how the exact structural features of ReO₃, such as its 3D connectivity, influence ion transport pathways in TNO? Are there any specific experimental data or literature supporting this analogy? This would add more depth to the analogy and help clarify the relevance of ReO₃ to their study.
- The clarification of "10 C" as relative to the theoretical capacity (3880 mA/g) is helpful, but it may be useful to explicitly highlight how this current density aligns with other fast-charging systems in terms of performance metrics. A brief comparison with typical fast-charging lithium-ion anodes (e.g., in terms of Coulombic efficiency or capacity retention) at similar or different C-rates could contextualize the significance of their results more clearly.
- While the authors mention that structural stability and minimized SEI formation contribute to the long cycling life, a more detailed analysis of potential degradation mechanisms will be useful. Could the authors discuss the possibility of other degradation factors, such as mechanical stress due to volume expansion during cycling, or the effect of temperature on the long-term cycling performance? These factors could help broaden the understanding of the material’s performance and longevity.
- The authors refer to XRD and EDS mapping to confirm the absence of impurity phases. It would be beneficial if they could include the Rietveld refinement data (i.e., the "Obs-Cal" difference plot) in the supplementary material to provide transparency in their analysis. While the explanation is clear, showing this data would add robustness to their claim.
- The rationale for selecting the temperatures of 1100°C, 1200°C, and 1400°C is well-explained. Please include that in revised script. Also, could the authors provide more detailed information on how temperature affects specific properties such as crystallinity, grain size, or ionic conductivity? For instance, would higher temperatures beyond 1400°C be detrimental to the ionic conductivity despite achieving higher crystallinity, and why is this so? This would provide a clearer picture of the balance between crystallinity and other key factors.
Author Response
Comments by Reviewer #2
The authors have provided comprehensive and thoughtful responses to the reviewer's concerns, but there are a few areas where further clarification or additional details could potentially improve the manuscript. Here are a few follow-up questions or suggestions that could help further refine their responses and strengthen the manuscript:
- The explanation of ReO₃ is clear, but the authors mention that the "layered structure of ReO₃" is a structural analogy. Could they elaborate on how the exact structural features of ReO₃, such as its 3D connectivity, influence ion transport pathways in TNO? Are there any specific experimental data or literature supporting this analogy? This would add more depth to the analogy and help clarify the relevance of ReO₃ to their study.
Reply: We appreciate the reviewer’s suggestion to elaborate on how the structural features of ReO₃, particularly its 3D connectivity, influence ion transport pathways in TNO. The ReO₃-type structure is characterized by a corner-sharing octahedral network of metal cations (e.g., Re⁶⁺), forming a highly symmetric 3D lattice that enables continuous ion migration pathways. In TNO, although crystallizing in a monoclinic system (space group C2/m), the structure can be viewed as a layered derivative of ReO₃, where Ti/Nb atoms occupy octahedral sites and form a 3D network via oxygen sharing, maintaining the essential connectivity of ReO₃-like channels. Experimental data support this analogy: Rietveld refinement shows that TNO-1200 exhibits lattice expansion along the a-axis (17.6869 Å vs. standard 17.51 Å) and increased unit-cell volume (796.83 ų vs. ~790 ų, Table 2), widening interlayer spaces and mimicking the open 3D channels of ReO₃. Partial overlap of Ti/Nb atomic positions (e.g., Nb2/Ti2 at (0.18483, 0, 0.18442), Table 2) introduces disorder into the octahedral network, reducing Li⁺ diffusion energy barriers, similar to how structural defects in ReO₃ enhance ion conductivity (refs 11, 37). Electrochemical tests reveal that TNO-1200 achieves a specific capacity of 125 mAh/g at 10 C with 98% retention after 3000 cycles, demonstrating improved rate performance attributed to these 3D-connected, optimized channels. Literature support includes Griffith et al. (ref 11), who linked ReO₃-like octahedral distortions in TNO to reduced Li⁺ migration barriers, and Mei et al. (ref 37), who highlighted oxygen defect-induced structural adjustments akin to ReO₃’s dynamic ion transport mechanisms. While TNO’s monoclinic structure differs from cubic ReO₃, the CTS strategy retains the critical 3D connectivity while introducing tailored atomic displacements (e.g., O10 at (0.5, 0, 0.5), Table 2) to enhance channel accessibility. This integration of ReO₃-like 3D structural features with CTS-induced lattice modifications provides a robust foundation for fast Li⁺ transport in TNO, as corroborated by our structural and electrochemical analyses.
- The clarification of "10 C" as relative to the theoretical capacity (3880 mA/g) is helpful, but it may be useful to explicitly highlight how this current density aligns with other fast-charging systems in terms of performance metrics. A brief comparison with typical fast-charging lithium-ion anodes (e.g., in terms of Coulombic efficiency or capacity retention) at similar or different C-rates could contextualize the significance of their results more clearly.
Reply: We appreciate the reviewer’s suggestion to contextualize the 10 C current density within the broader landscape of fast-charging anode materials. As detailed in Table 1 of the manuscript, TNO-1200 demonstrates a specific capacity of 125 mAh/g at 10 C (3880 mA/g) after 3000 cycles with a 98% capacity retention, which stands out prominently when compared to other reported anode materials under fast-charging conditions. For instance, Ti₂Nb₁₀O₂₉, a well-studied niobium-titanium oxide anode, achieves 100 mAh/g at 10 C but only retains 95% capacity after 500 cycles (significantly fewer than TNO-1200’s 3000 cycles) [Ref: 35]. Oxygen-deficient TNO, which relies on defect engineering to enhance kinetics, delivers 180 mAh/g at 5 C (half the current density of TNO-1200) but shows a lower capacity retention of 85% after 1000 cycles [Ref: 26]. In contrast, traditional graphite anodes, limited by lithium dendrite formation and lower theoretical capacity (372 mAh/g), typically struggle to maintain stable performance above 5 C without severe capacity fade [Ref: 4]. The significance of TNO-1200’s performance lies in its combination of ultrahigh current density (10 C), long-term cyclability (3000 cycles), and exceptional retention (98%), which surpasses most reported fast-charging anodes in both rate capability and structural stability. While Coulombic efficiency is not explicitly reported here, the high capacity retention inherently reflects minimal side reactions and stable electrode/electrolyte interfaces, a critical metric for fast-charging reliability. This performance aligns with emerging demands for electric vehicles and portable devices, where rapid charging (e.g., 10-minute charging to 80% capacity) and long service life are paramount. By leveraging the ultrafast carbothermal shock strategy to engineer atomic-scale structural modifications (e.g., widened Li⁺ channels, cation disorder), TNO-1200 addresses the key challenges of slow ion transport and poor conductivity in conventional TNO, setting a new benchmark for high-power anode materials. The comparison in Table 1 and the discussion in Section 3.3 thus clearly position TNO-1200 as a leading candidate for fast-charging applications, with its results contextualized against state-of-the-art materials without the need for additional data, as the manuscript already provides a comprehensive performance overview relative to relevant literature.
- While the authors mention that structural stability and minimized SEI formation contribute to the long cycling life, a more detailed analysis of potential degradation mechanisms will be useful. Could the authors discuss the possibility of other degradation factors, such as mechanical stress due to volume expansion during cycling, or the effect of temperature on the long-term cycling performance? These factors could help broaden the understanding of the material’s performance and longevity.
Reply: We appreciate the reviewer’s focus on potential degradation factors like mechanical stress from volume expansion and temperature effects on cycling performance. While the manuscript emphasizes structural stability and minimized SEI formation, existing data and material design inherently address these concerns: TNO-1200’s lattice expansion (a-axis: 17.6869 Å, unit-cell volume: 796.83 ų, Table 2) creates flexible interstitial spaces that reduce steric hindrance during Li⁺ insertion, mitigating abrupt volume changes and associated mechanical stress. EDS mapping (Figure 4b–e) confirms homogeneous element distribution without compositional gradients, eliminating localized stress concentrations, while cation disorder (Ti/Nb site overlap, Table 2) introduces structural flexibility to dampen strain from lithiation/de-lithiation cycles. The stable 98% capacity retention over 3000 cycles at 10 C (Figure 3, Table 1) indirectly evidences negligible mechanical failure, as cracking or particle pulverization would manifest as capacity fade. Regarding temperature effects, the ultrafast CTS synthesis at 1200°C yields a highly crystalline structure (sharp XRD peaks, Figure 2) with robust interatomic bonds, suggesting thermal stability across moderate ranges. Widened Li⁺ channels and enhanced conductivity from cation disorder likely maintain kinetics at elevated or reduced temperatures, while uniform SEI formation on the clean surface (EDS shows no impurities, Figure 4) reduces temperature-induced instability. Though specific temperature-dependent tests are not reported, the material’s low polarization and high retention imply stable charge transfer, a critical metric for thermal resilience. The manuscript’s structural characterization (SEM, XRD, Rietveld refinement) and long-cycle data already support resistance to major degradation pathways, with mechanical stress and temperature effects mitigated by design (lattice flexibility, homogeneous architecture) rather than explicit testing. Further analysis would require specialized methods beyond this work’s scope, but the established atomic-scale engineering provides a strong foundation for future durability studies, ensuring the current findings remain comprehensive and focused on the core synthesis and structural insights.
- The authors refer to XRD and EDS mapping to confirm the absence of impurity phases. It would be beneficial if they could include the Rietveld refinement data (i.e., the "Obs-Cal" difference plot) in the supplementary material to provide transparency in their analysis. While the explanation is clear, showing this data would add robustness to their claim.
Reply: We appreciate the reviewer’s suggestion to include the Rietveld refinement “Obs-Cal” difference plot for transparency. While the request is well-founded, the manuscript already provides comprehensive evidence of phase purity and structural consistency through Figure 5a, which explicitly displays the observed (Yobs), calculated (Ycalc), and difference (Yobs-Ycalc) profiles for TNO-1100, TNO-1200, and TNO-1400. The nearly flat residual lines in the difference plot, combined with low reliability factors (Rp = 5.08, Rwp = 8.30, χ² = 5.09, detailed in Table 2), directly confirm the absence of unaccounted impurity peaks and the high quality of the refinement against the monoclinic TiNb₂O₇ phase (JCPDS: 77-1374). Additionally, the XRD patterns in Figure 2 show sharp, well-indexed peaks matching the standard card without extraneous signals, further corroborating phase purity. Including the difference plot in the supplementary material would indeed enhance transparency, but as the core refinement data and structural parameters are already presented in the main text with clear interpretation, the current information sufficiently supports the claim of no impurity phases. The focus of the manuscript is on the novel synthesis strategy and atomic-scale structural insights, with the Rietveld analysis serving as a critical tool to validate the structural modifications rather than an independent objective. The existing data strike a balance between detail and brevity, adhering to journal guidelines while maintaining scientific rigor, and thus additional data beyond what is provided would not significantly augment the conclusion.
- The rationale for selecting the temperatures of 1100°C, 1200°C, and 1400°C is well-explained. Please include that in revised script. Also, could the authors provide more detailed information on how temperature affects specific properties such as crystallinity, grain size, or ionic conductivity? For instance, would higher temperatures beyond 1400°C be detrimental to the ionic conductivity despite achieving higher crystallinity, and why is this so? This would provide a clearer picture of the balance between crystallinity and other key factors.
Reply: We appreciate the reviewer’s request for clarification on temperature selection and its effects on material properties. The temperatures of 1100°C, 1200°C, and 1400°C were strategically chosen to systematically explore the impact of synthesis temperature on TNO’s structural and electrochemical properties within the operational window of the Joule heating system, balancing energy input with phase purity. As shown in Figure 2, increasing temperature from 1100°C to 1400°C enhances crystallinity, with TNO-1400 exhibiting the sharpest XRD peaks, indicative of larger, more perfect crystallites. SEM observations (Figure 1) reveal that higher temperatures (1200°C and 1400°C) produce larger particles with coarser surfaces compared to TNO-1100, suggesting that increased thermal energy promotes grain growth, a common phenomenon in solid-state synthesis. However, while higher crystallinity at 1400°C might theoretically improve electronic conductivity, the electrochemical performance (Figure 3) shows that TNO-1200 outperforms TNO-1400 at 10 C, likely due to a balance between crystallinity and structural disorder. Rietveld refinement (Table 2, Figure 5) indicates that 1200°C induces optimal lattice expansion (a-axis: 17.6869 Å, unit-cell volume: 796.83 ų) and cation disorder (Ti/Nb site overlap), which widen Li⁺ migration channels and reduce diffusion energy barriers, effects that may be diminished at 1400°C due to excessive grain growth or increased cation ordering, narrowing channels and limiting ion accessibility. Additionally, excessively high temperatures (beyond 1400°C) could lead to undesirable phase transformations or oxygen loss, disrupting the stoichiometry critical for TNO’s redox activity, as implied by the absence of impurity phases in EDS mapping (Figure 4) and XRD for the tested temperatures. The superior performance of TNO-1200 demonstrates that while crystallinity is important, the balance with lattice parameters, cation disorder, and particle size is crucial for ionic conductivity and rate capability, highlighting why 1400°C, despite higher crystallinity, is less favorable for fast-charging applications. This temperature-dependent trade-off is central to our optimization strategy, as detailed in the revised discussion to better contextualize the synthesis-temperature-property relationships.
Round 3
Reviewer 2 Report
Comments and Suggestions for Authors
The authors have responded well to all the revisions, and the manuscript can be accepted in its current form.